# Alpha kinase 1 controls intestinal inflammation by suppressing the IL-12/Th1 axis

Grigory Ryzhakov[1], Nathaniel R. West[1,6], Fanny Franchini[1], Simon Clare[2], Nicholas E. Ilott[1], Stephen N. Sansom[1], Samuel J. Bullers[1], Claire Pearson[1], Alice Costain[1], Alun Vaughan-Jackson[1], Jeremy A. Goettel[3], Joerg Ermann [4], Bruce H. Horwitz[4], Ludovico Buti[5], Xin Lu [5], Subhankar Mukhopadhyay[2], Scott B. Snapper[3] & Fiona Powrie[1]

Inflammatory bowel disease (IBD) are heterogenous disorders of the gastrointestinal tract caused by a spectrum of genetic and environmental factors. In mice, overlapping regions of chromosome 3 have been associated with susceptibility to IBD-like pathology, including a locus called *Hiccs*. However, the specific gene that controls disease susceptibility remains unknown. Here we identify a *Hiccs* locus gene, *Alpk1* (encoding alpha kinase 1), as a potent regulator of intestinal inflammation. In response to infection with the commensal pathobiont *Helicobacter hepaticus* (*Hh*), Alpk1-deficient mice display exacerbated interleukin (IL)-12/IL-23 dependent colitis characterized by an enhanced Th1/interferon(IFN)-γ response. Alpk1 controls intestinal immunity via the hematopoietic system and is highly expressed by mononuclear phagocytes. In response to *Hh*, Alpk1$^{-/-}$ macrophages produce abnormally high amounts of IL-12, but not IL-23. This study demonstrates that Alpk1 promotes intestinal homoeostasis by regulating the balance of type 1/type 17 immunity following microbial challenge.

[1] Kennedy Institute of Rheumatology, University of Oxford, Oxford OX3 7FY, United Kingdom. [2] Wellcome Trust Sanger Institute, Hinxton, Cambridge CB10 1SA, United Kingdom. [3] Boston Children's Hospital and Harvard Medical School, Boston, MA 02115, USA. [4] Department of Gastroenterology, Brigham and Women's Hospital and Harvard Medical School, Boston, MA 02115, USA. [5] Ludwig Institute of Cancer Research, University of Oxford, Oxford OX3 7DQ, United Kingdom. [6] Present address: Genentech, Department of Cancer Immunology, South San Francisco, CA 94080, USA. These authors contributed equally: Grigory Ryzhakov, Nathaniel R West. Correspondence and requests for materials should be addressed to F.P. (email: fiona.powrie@kennedy.ox.ac.uk)

Inflammatory bowel disease (IBD) pathogenesis is mechanistically complex and includes elements of genetic susceptibility, immune dysregulation, environmental factors, and the microbiome. As with humans, colitis in mice is strongly influenced by host genetics, such that different inbred strains exhibit widely divergent phenotypes in models of IBD[1,2]. For example, whereas $129SvEv.Rag2^{-/-}$ mice develop aggressive colitis following infection with $Hh$, $C57BL/6.Rag1^{-/-}$ mice do not[3]. $129SvEv$ mice deficient for the Wiskott-Aldrich syndrome protein ($Was^{-/-}$) are also susceptible to spontaneous colitis[4].

In recent years, two loci on chromosome 3—termed "cytokine-dependent colitis susceptibility locus" ($Cdcs1$) in C3H/HeJBir mice, and "Helicobacter hepaticus-induced colitis and associated cancer susceptibility" ($Hiccs$) in 129SvEv mice —have been identified as the critical genetic determinants of colitis susceptibility in these strains[2,3,5–8]. The $Cdcs1$ locus was first identified in the context of interleukin-10 deficient ($Il10^{-/-}$) mice over 10 years ago[5,8], and also controls spontaneous colitis in $C57BL/6.Tbx21^{-/-}Rag2^{-/-}$ (TRUC) mice[9]. The $Hiccs$ locus, located in a similar region of chromosome 3, controls susceptibility to $Hh$-induced colitis and eventual onset of colitis-associated colorectal cancer[3]. The ability of the $Cdcs1$ and $Hiccs$ loci to control colitis susceptibility in several mechanistically distinct models suggests that they include one or more critical immunoregulatory genes. However, individual $Cdcs1/Hiccs$-related genes have not been directly studied in the context of intestinal inflammation.

A $Hiccs$ locus gene, Alpha kinase 1 ($Alpk1$), has multiple non-synonymous single nucleotide differences in its coding region between the colitis-susceptible 129SvEv and the resistant C57BL/6 mouse strains[3]. We previously showed that $Alpk1$ expression is upregulated in response to inflammatory stimuli in myeloid cells[3]. Also, single nucleotide polymorphisms (SNPs) in the human $ALPK1$ gene have been linked to a variety of inflammatory conditions, including gout and chronic kidney disease[10,11]. More recently, in vitro studies suggested that Alpk1 mediates pathogen-induced IL-8 expression in gastric epithelial cells[12,13], making it a relevant gene to explore in the context of gut inflammation.

In this study, we further refine the $Cdcs1$ locus and identify a core colitis-determining region that is essentially identical to the $Hiccs$ locus. To address a potential role for Alpk1 in regulation of intestinal homoeostasis, we have generated Alpk1-deficient mice. We show that Alpk1 deficiency leads to severe colitis and an exaggerated Th1 immune response in mice infected with the intestinal pathobiont Helicobacter hepaticus. We further demonstrate that Alpk1 exerts its anti-inflammatory function within the hematopoietic compartment, in which it restrains production of IL-12 by myeloid cells in response to Helicobacter challenge.

## Results

**A genetic locus controlling colitis susceptibility in mice.** In addition to the $Il10^{-/-}$ and TRUC models, we now show that the risk-conferring genotype of $Cdcs1$ also confers susceptibility to spontaneous colitis in $C57BL/6.Was^{-/-}$ mice (Supplementary Fig. 1a–c). In both $Was^{-/-}$ and TRUC mice, colitis susceptibility requires homozygosity for the C3H-derived $Cdcs1$ allele in hematopoietic cells (Supplementary Fig. 1d–e)[9]. Previously, we mapped the critical region of the $Hiccs$ locus to a 1.71-Mb interval that contains five microRNAs and eight protein-coding genes[3]. We have similarly fine-mapped the susceptibility-controlling region of $Cdcs1$ in TRUC mice to a congenic interval flanked by the genetic markers $D3Mit348$ and $D3Mit319$ (Supplementary Fig. 2). This $Cdcs1$ core region is essentially identical with $Hiccs$ (Fig. 1a). Therefore, we hypothesized that this locus contains a previously unidentified gene that controls colitis susceptibility in multiple mouse models of IBD.

**Early IFN-γ response in $Hh$-treated colitis-susceptible mice.** A congenic 129SvEv strain carrying the B6 allele of the $Hiccs$ locus ($129.Hiccs^{B6}.Rag2^{-/-}$) phenocopies the colitis resistance displayed by C57BL/6 mice (Fig. 1b)[3]. To better understand how the $Hiccs$ locus regulates colitis, we performed transcriptomic analysis of colon tissue from $129.Rag2^{-/-}$ and $129.Hiccs^{B6}.Rag2^{-/-}$ mice during the early phase of $Hh$ infection. As early as 2 days post infection, we observed robust induction of inflammatory genes, many of which are known to be interferon-regulated, in $129.Rag2^{-/-}$ but not $129.Hiccs^{B6}.Rag2^{-/-}$ mice (Fig. 1c, Supplementary Data 1). Indeed, independent gene set enrichment (GSEA) and gene ontology (GO) analyses identified IFN-γ-response genes as being highly enriched in $Hh$-infected $129.Rag2^{-/-}$ mice (Fig. 1d, e). IFN-γ was highly overexpressed in colon tissue of $Hh$-infected $129.Rag2^{-/-}$ mice at both the mRNA and protein level (Fig. 1f, g). The genes encoding the p35 and p40 subunits of IL-12 ($Il12a$ and $Il12b$), a critical upstream driver of IFN-γ expression, were similarly overexpressed by $129.Rag2^{-/-}$ mice (Fig. 1f).

**Alpk1 deficiency confers susceptibility to innate colitis.** The rapid kinetics of $Hh$-induced inflammation in $129.Rag2^{-/-}$ mice suggest that the $Hiccs$ locus may control some aspect of acute bacterial recognition and effector function. Among $Hiccs$ locus genes, $Alpk1$ and $Tifa$ have been implicated in bacterial recognition by epithelial cells[12–15]. Interestingly, while the 129 and B6 alleles of $Alpk1$ differ by 17 non-synonymous polymorphisms, no such differences exist in $Tifa$[3]. Therefore, we generated $Alpk1$ loss-of-function mice on the C57BL/6 background ($Alpk1^{-/-}$) via CRISPR/Cas9-mediated disruption of $Alpk1$ exon 10. At steady state, $Alpk1^{-/-}$ mice were viable, healthy, and showed no obvious immunological phenotypes compared to wild type or heterozygous littermates[16]. $Alpk1^{-/-}$ mice were then bred with $C57BL/6\ Rag1^{-/-}$ mice to be tested in the $Hh$-induced innate colitis model (Fig. 2a). $Hh$-infected $Alpk1^{-/-}Rag1^{-/-}$ animals developed severe inflammation in the caecum and colon, whereas $Alpk1^{+/-}Rag1^{-/-}$ mice developed mild inflammation (Fig. 2b, c), with the largest differences between genotypes occurring in the mid- and distal colon regions (Fig. 2d). Thus, Alpk1 deficiency causes a colitis-susceptible phenotype similar to that of $129.Rag2^{-/-}$ mice, suggesting $Alpk1$ as a potential regulator of colitis susceptibility in the $Hiccs$ and $Cdcs1$ loci.

In agreement with colon histopathology, total numbers of colonic $CD45^+$ leucocytes, particularly neutrophils, monocytes, and $Ly6C^+$ macrophages, were significantly higher in $Hh$-infected $Alpk1^{-/-}$ versus $Alpk1^{+/-}$ animals, as was the frequency of TNFα expression among myeloid cells (Fig. 2e–f, Supplementary Fig. 3a–d). Our previous study demonstrated that innate lymphoid cells (ILCs) are critical for $Hh$-induced inflammation in $129.Rag2^{-/-}$ mice[3,17]. Although, the IFN-γ$^+$ ILC frequency in colon tissue from $Alpk1^{+/-}$ and $Alpk1^{-/-}$ littermates was similar (Fig. 2e, f), the intensity of IFN-γ expression in $Alpk1^{-/-}$ ILCs was significantly increased (Fig. 2e, f). By contrast, production of IL-22 and IL-17A by colonic $Alpk1^{-/-}$ ILCs was unaltered or reduced (Supplementary Fig. 3e–g). Whole-colon tissue of colitic $Alpk1^{-/-}Rag1^{-/-}$ mice showed high expression of $Ifng$, $Il12a$, $Il12b$, $Il23a$, and other inflammatory factors compared to heterozygous littermates (Fig. 2g). We also observed elevated $Il12b$ mRNA expression in the colons of $Hh$-infected $Alpk1^{-/-}Rag1^{-/-}$ mice using RNAScope in situ hybridization (Supplementary Fig. 3h)[18]. As with $Hh$-infected $129.Rag2^{-/-}$ mice[19–21], treatment with an IL-12p40 neutralising antibody, which targets both IL-12 and IL-23, strongly suppressed colonic inflammation in $Alpk1^{-/-}Rag1^{-/-}$ mice, while blocking the

IL-23 receptor (IL-23R) only partially reduced pathology (Fig. 2h). Therefore, both IL-12 and IL-23 are non-redundant drivers of *Hh*-induced colitis in *Alpk1*$^{-/-}$*Rag1*$^{-/-}$ mice.

**The role of Alpk1 in a lymphocyte-replete model of colitis.** Having found a critical role for Alpk1 in regulating inflammation in a *Rag1*-deficient setting, we next assessed its impact on lymphocyte-replete colitis. *Hh* infection does not induce colitis in wild-type B6 mice unless IL-10 signalling is blocked[22]. We

therefore treated *Alpk1*$^{+/-}$ and *Alpk1*$^{-/-}$ mice with *Hh* and an IL-10R (IL-10 receptor alpha) blocking antibody to induce colitis (Fig. 3a). In agreement with results obtained using *Rag1*$^{-/-}$ mice, *Alpk1*$^{-/-}$ animals developed exacerbated intestinal inflammation based on histological scoring (Fig. 3b, c), colonoscopy (Fig. 3d), and total CD45$^{+}$ lamina propria leucocytes (Fig. 3e). Interestingly, only complete Alpk1 deficiency resulted in exacerbated pathology, as heterozygous (*Alpk1*$^{+/-}$) and wild-type (*Alpk1*$^{+/+}$) littermates displayed comparably modest colitis (Supplementary

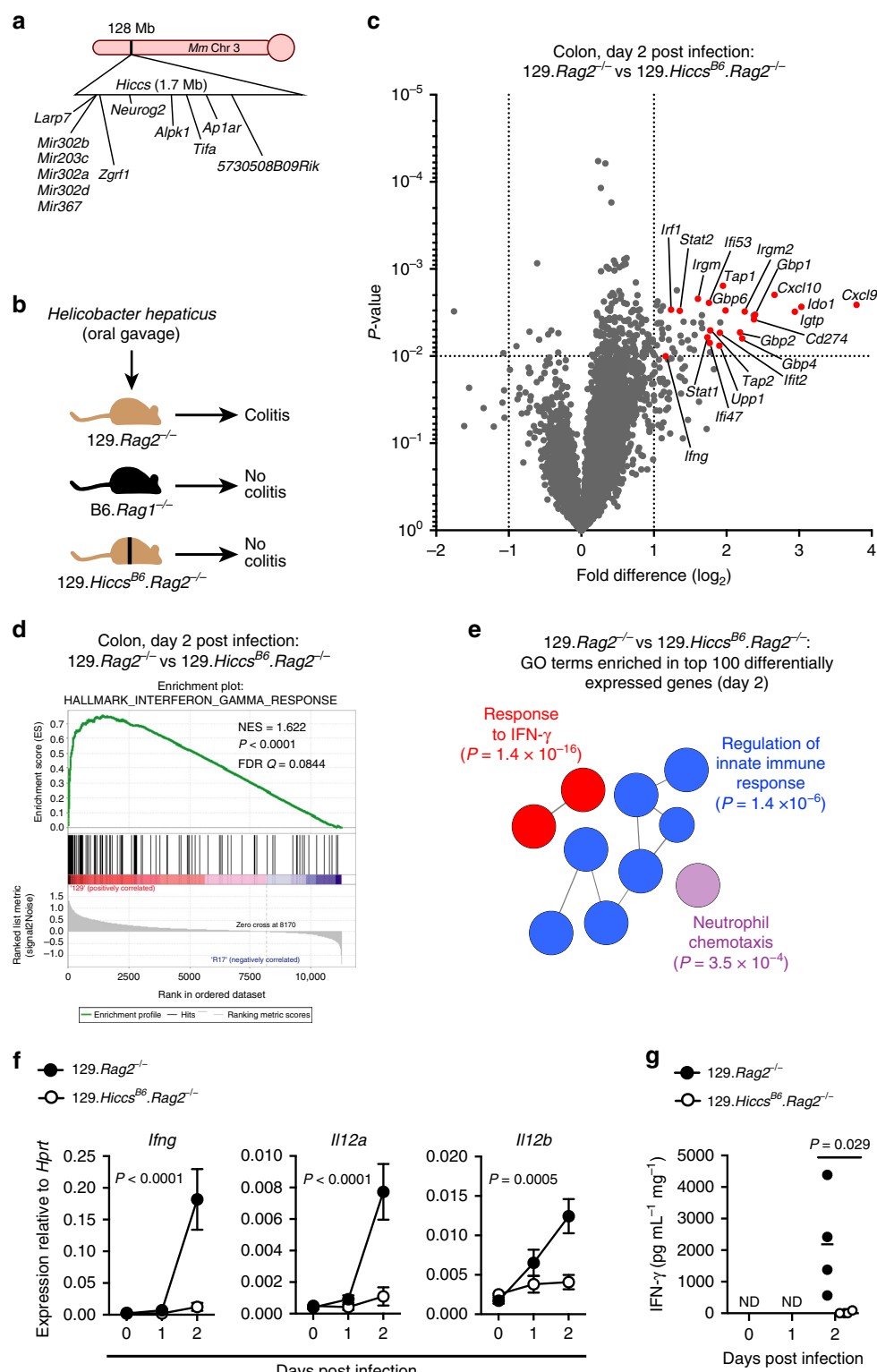

Fig. 4a). Consistent with their increased colitis, $Alpk1^{-/-}$ mice also displayed moderate splenomegaly (Supplementary Fig. 4b). The increased disease burden in $Alpk1^{-/-}$ mice was not due to increased intestinal colonization by $Hh$ (Supplementary Fig. 4c).

The $Hh$ + anti-IL10R protocol normally elicits a $CD4^+$ helper T-cell response with mixed Th1 and Th17 characteristics. However, compared to WT or heterozygous controls, intestinal $CD4^+$ T cells of $Alpk1^{-/-}$ mice were highly skewed towards a Th1 phenotype ($IFN$-$\gamma^+$ $IL$-$17A^-$) following induction of colitis, as measured by cell frequency and $IFN$-$\gamma$ expression per cell (Fig. 3f, g, Supplementary Fig. 4d, e). In keeping with their similar degree of colitis, wild type and $Alpk1^{+/-}$ mice showed comparable T cell phenotypes; only $Alpk1^{-/-}$ mice developed a polarized Th1 response (Supplementary Fig. 4d). Although $IFN$-$\gamma$ was largely co-expressed with $TNF$-$\alpha$, frequencies of $TNF$-$\alpha$ expression were not significantly elevated in $Alpk1^{-/-}$ T cells (Supplementary Fig. 4f). In contrast, the frequency of Th17 cells ($IFN$-$\gamma^-$ $IL$-$17A^+$) and their intensity of $IL$-$17A$ expression was diminished in $Alpk1^{-/-}$ mice (Fig. 3f, g, Supplementary Fig. 4e). $IL$-$22$ producing $CD4^+$ T cells were also less frequent in $Alpk1^{-/-}$ mice (Supplementary Fig. 4g). Boolean gating analysis of cytokine-producing $CD4^+$ T cells confirmed a profound shift in effector phenotype in the absence of Alpk1, with $IFN$-$\gamma^+IL$-$17A^-$ cells vastly outnumbering other populations (Supplementary Fig. 4h). Although the total number of colonic $FOXP3^+$ regulatory T cells (Treg) was comparable between $Alpk1^{-/-}$ and control mice, their frequency among $CD4^+$ T cells was reduced in $Alpk1^{-/-}$ mice, particularly the $FOXP3^+ROR\gamma t^+$ subset that is induced in response to $Hh$ (Supplementary Fig. 4i)[23]. While we observed a similarly Th1-skewed phenotype in the mesenteric lymph nodes of $Alpk1^{-/-}$ animals, this was not apparent in their peripheral lymph nodes (inguinal/axillary) or spleens (Supplementary Fig. 4j), suggesting that Alpk1 exerts local control over Th1 immunity.

Gene expression analysis of whole-colon tissue confirmed elevated amounts of pro-inflammatory and Th1 cytokines in colitic $Alpk1^{-/-}$ mice, while expression of Th2 cytokines and $IL$-$10$ were reduced relative to WT/heterozygous animals (Supplementary Fig. 4k). Interestingly, $Alpk1^{-/-}$ mice displayed a robust intestinal Th1 response and reduced frequency of $FOXP3^+ROR\gamma t^+$ Treg following $Hh$ infection in the absence of IL-10R blockade (Fig. 3h–k). This alteration in Th1/Treg balance did not cause marked pathology or recruitment of myeloid leucocytes to the colon, however (Fig. 3h, Supplementary Fig. 4l). In contrast, $Alpk1^{-/-}$ mice treated with anti-IL-10R antibody alone did not show any signs of pathology or activation of a Th1 response (Fig. 3h, i, k). Therefore, Alpk1 deficiency augments the IL-12/Th1 axis and reduces the Treg/Th1 ratio following $Hh$ infection, causing mild immunopathology that is exacerbated in the absence of IL-10 signalling.

**Alpk1 controls colitis severity via hematopoietic cells.** Analysis of the Immgen database[24] and FACS-sorted intestinal populations revealed a broad spectrum of Alpk1-expressing cell types, including antigen presenting cells (APCs), epithelial cells, and intestinal stromal cells (Supplementary Fig. 5). To determine whether Alpk1 impacts intestinal inflammation via the hematopoietic compartment, we conducted reciprocal bone marrow chimera experiments. Consistent with prior observations of the $Hiccs$ and $Cdcs1$ loci[3] (Supplementary Fig. 1-2), irradiated WT mice reconstituted with $Alpk1^{-/-}$ bone marrow developed more severe $Hh$ + anti-IL-10R-induced colitis in comparison to those that received WT bone marrow (Fig. 4a–c). Conversely, irradiated WT or $Alpk1^{-/-}$ animals reconstituted with wild-type bone marrow developed comparable levels of colitis (Fig. 4d–f), demonstrating that Alpk1 regulates colitis via the hematopoietic compartment.

Because $CD4^+$ T-cell responses are largely programmed by APCs, we employed the naive T-cell transfer model of colitis (i.e., adoptive transfer of naive $CD4^+$ T cells to $Rag1^{-/-}$ hosts) to test whether, in the absence of acute bacterial challenge, deficiency of Alpk1 in APCs could influence the phenotype of expanding T cells. However, transfer of WT T cells to $Alpk1^{-/-}Rag1^{-/-}$ or $Alpk1^{+/-}Rag1^{-/-}$ hosts resulted in equivalent T-cell differentiation and severity of colitis (Supplementary Fig. 6a–d). Similarly, no differences were observed between $Alpk1^{+/+}Rag1^{-/-}$ recipients following transfer of $Alpk1^{-/-}$ or WT T cells (Supplementary Fig. 6e–h). Thus, Alpk1 deficiency in the absence of a strong microbial driver does not appear to significantly impact $CD4^+$ T-cell differentiation or susceptibility to T-cell-driven colitis. We also tested the role of Alpk1 in another model of colitis that bypasses bacterial stimuli, in which treatment of $Rag1^{-/-}$ mice with an agonistic anti-CD40 antibody directly activates APCs and induces IL-12/IL-23-mediated pathology[21]. $Alpk1^{+/-}Rag1^{-/-}$ and $Alpk1^{-/-}Rag1^{-/-}$ mice developed comparable levels of inflammation following anti-CD40 challenge (Supplementary Fig. 6i–n), providing further evidence that Alpk1 is functionally linked to microbial sensing.

**Alpk1 regulates $Hh$-driven IL-12 production by phagocytes.** Alpk1 has been proposed to affect several distinct cellular pathways[13,25–29], yet none explain the pathological effect of its deficiency in the gut. However, our in vivo data suggest that the defect may be related to bacterially induced IL-12 production. Because APCs are the major producers of IL-12, we generated bone marrow-derived macrophages (BMDMs) via culture of bone marrow with GM-CSF (granulocyte-macrophage colony stimulating factor) and examined their response to $Hh$ (Fig. 5a). After 8 days of culture, flow cytometry analysis revealed no clear differences in differentiation between WT and $Alpk1^{-/-}$ cells (Supplementary Fig. 7a–c). Compared to WT cells, $Alpk1^{-/-}$ BMDMs secreted more IL-12 following exposure to $Hh$, whereas IL-23 production was similar (Fig. 5b). At the mRNA level, expression of $Il12a$ and $Il12b$, but not $Il23a$, was significantly

**Fig. 1** $129.Rag2^{-/-}$ mice rapidly activate interferon-gamma signaling following infection with $Hh$. **a** Schematic of the $Hiccs$ locus, which regulates sensitivity to $Hh$-driven innate colitis. **b** Colitis phenotypes after $Hh$ infection of $B6.Rag1^{-/-}$, $129.Rag2^{-/-}$, or congenic $129.Rag2^{-/-}$ mice bearing a $Hiccs$ locus that matches the B6 genotype ($129.Hiccs^{B6}.Rag2^{-/-}$). **c–e** Transcriptomic profiling of whole-colon tissue from $129.Rag2^{-/-}$ and $129.Hiccs^{B6}.Rag2^{-/-}$ mice 2 days after oral $Hh$ infection ($n = 4$ per group). **c** Tornado plot displaying $\log_2$ fold differences in $129.Rag2^{-/-}$ versus $129.Hiccs^{B6}.Rag2^{-/-}$ mice. IFN-$\gamma$-regulated genes are indicated in red. **d** Gene set enrichment analysis (GSEA), showing enrichment of the "interferon-gamma response" hallmark gene set in colon tissue from $129.Rag2^{-/-}$ mice. **e** The top 100 differentially expressed genes (high in $129.Rag2^{-/-}$ mice) were analyzed for gene ontology (GO) term enrichment using ClueGO. The most highly enriched GO terms are shown, with lead terms and associated $p$-values indicated. **f** RT-qPCR analysis of whole-colon gene expression after $Hh$ infection. Data analyzed by two-way ANOVA ($n = 4$ per group per time point). $P$-values indicate day 2 comparisons corrected with Sidak's multiple comparisons tests (DF = 17 for each comparison). **g** IFN-$\gamma$ secretion by colon explant tissue cultured overnight ex vivo at steady state or following $Hh$ infection. ND no IFN-$\gamma$ detected in culture supernatant. Data ($Hh$ day 2) analyzed by Mann–Whitney test ($n = 4$ per group/ timepoint). Data in **c–g** are from one experiment

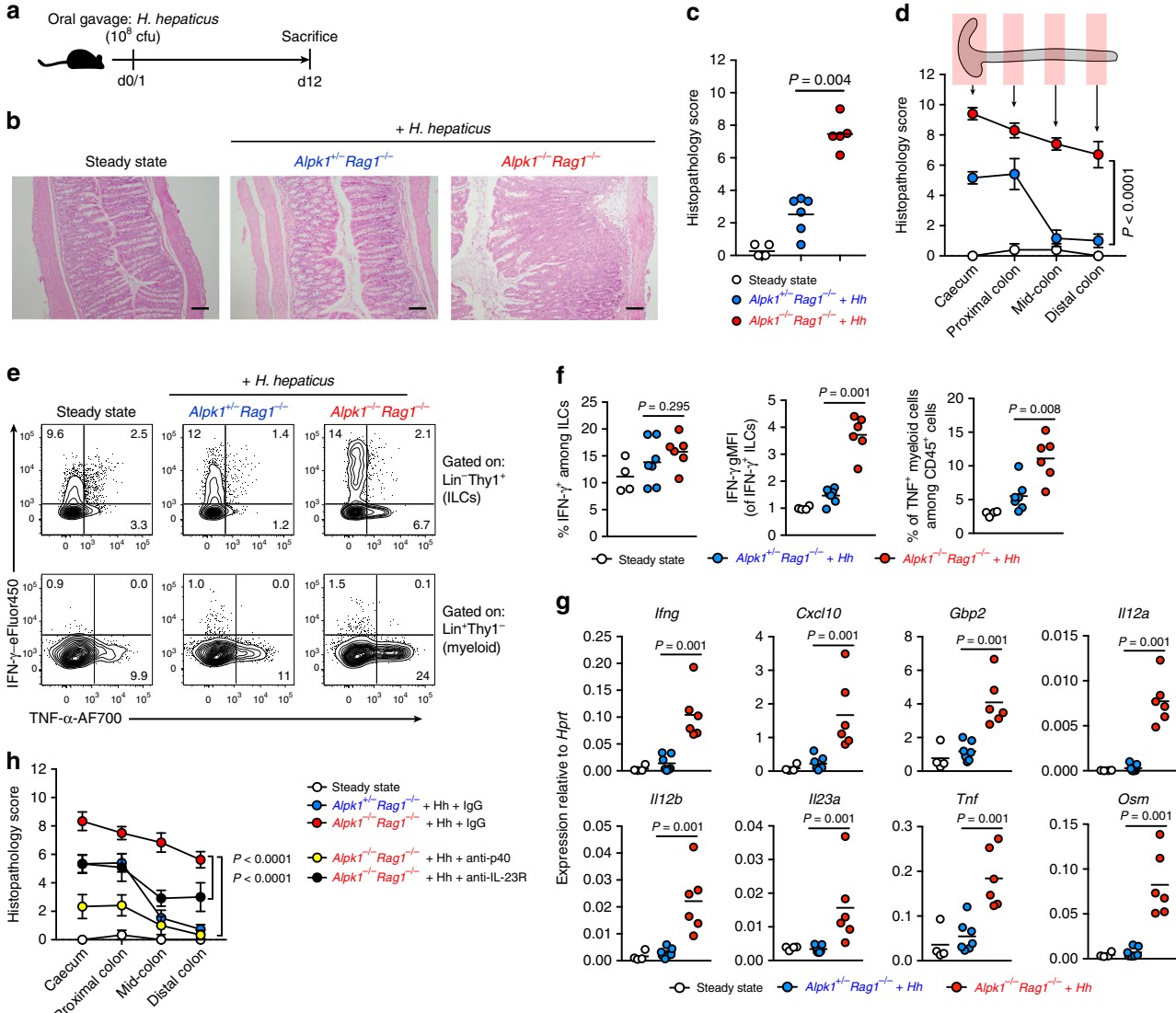

**Fig. 2** Alpk1-deficient mice are highly susceptible to *Hh*-induced innate colitis. **a** Experimental schematic of *Hh* infection using B6.*Rag1*[−/−]*Alpk1*[−/−] mice or B6.*Rag1*[−/−]*Alpk1*[+/−] littermate controls. **b** Representative H&E stained mid-colon cross-sections at steady state and after 12 days of *Hh* infection. Scale bars = 100 μm. **c–d** Overall colon histopathology scores (**c**) and histopathology scores along the length of the large intestine, from caecum to distal colon (**d**). Steady state (*n* = 5), B6.*Rag1*[−/−]*Alpk1*[−/−] + *Hh* (*n* = 5) and B6.*Rag1*[−/−]*Alpk1*[+/−] + *Hh* (*n* = 6). Data shown are from one of three representative experiments and only the *Hh*-treated groups were analyzed by Mann–Whitney test (**c**) or by two-way ANOVA (genotypes of the *Hh* infected groups compared, corrected with Sidak's multiple comparisons test, DF = 52) (**d**). **e–f** Flow cytometry analysis of cytokine expression by colon lymphocyte and myeloid populations after PMA/ionomycin restimulation. Frequency of IFN-γ expression among ILCs, geometric mean fluorescence intensity (gMFI) of IFN-γ[+] cells, and frequency of TNF expression among myeloid cells are shown in **f**. Only the *Hh*-treated groups were taken into analysis by Mann–Whitney test. **g** Whole-colon expression of *Ifng*, IFN-γ-regulated genes, and selected inflammatory cytokines analyzed by RT-qPCR. Only the *Hh*-treated groups were analyzed by Mann–Whitney test. **e–g** Data summarize two independent experiments: steady state (*n* = 4), B6.*Rag1*[−/−]*Alpk1*[−/−] + *Hh* (*n* = 6) and B6.*Rag1*[−/−]*Alpk1*[+/−] + *Hh* (*n* = 7). **h** Colon histopathology scores from steady-state mice (*n* = 6), *Hh*-infected B6.*Rag1*[−/−]*Alpk1*[+/−] mice (*n* = 15), and B6.*Rag1*[−/−]*Alpk1*[−/−] animals infected with *Hh* and treated with IgG isotype control antibodies (*n* = 12), IL-12p40 neutralizing antibodies (*n* = 6), or IL-23R neutralizing antibodies (*n* = 6). Data were pooled from two independent experiments and analyzed by two-way ANOVA. *P*-values indicate treatment comparisons (B6.*Rag1*[−/−]*Alpk1*[−/−] + *Hh* IgG vs anti-p40 or IgG vs anti-IL23R) corrected with Sidak's multiple comparisons tests (DF = 159 for each comparison)

increased in *Alpk1*[−/−] BMDMs (Fig. 5c). IL-12 secretion by *Hh* was blocked in the presence of anti-TLR2 antibody (Fig. 5b), suggesting that Alpk1 mediates signal transduction downstream of TLR2 and upstream of *Il12a*/*Il12b* transcription. Interestingly, pure TLR2 and TLR4 agonists, such as Pam3CSK4 and LPS-induced comparable levels of IL-12 in both wild type and *Alpk1*[−/−] cells (Fig. 5b).

**Transcriptomic analysis of Alpk1-regulated genes**. To examine the role of Alpk1 in an unbiased manner, we compared whole transcriptomes of *Hh*-treated wild type and *Alpk1*[−/−] BMDMs using RNA sequencing. Under resting conditions, the transcriptomes of *Alpk1*[−/−] and *Alpk1*[+/−] cells were almost indistinguishable by tSNE (t-stochastic neighbour embedding) analysis (Supplementary Fig. 8a), with 32 genes (including *Alpk1* itself)

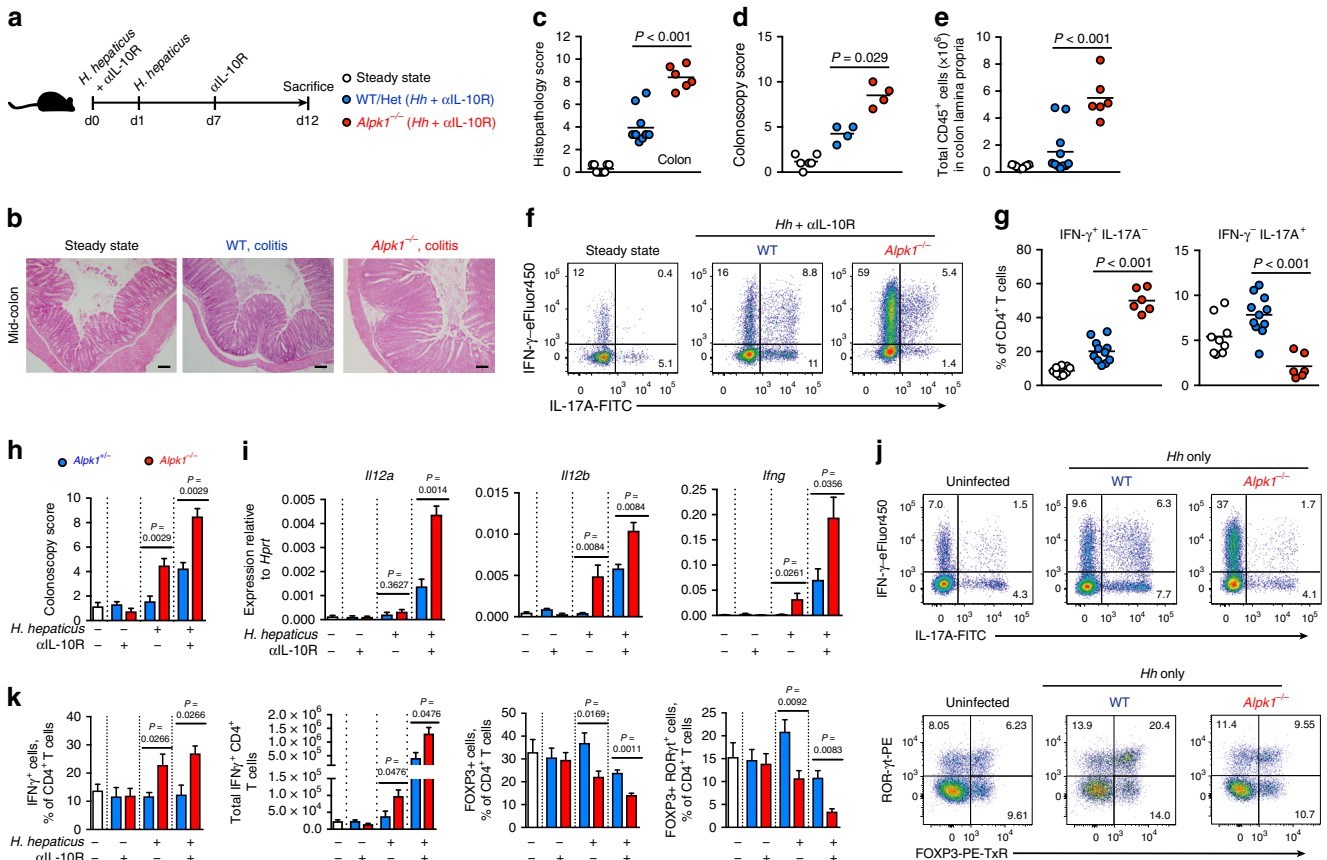

**Fig. 3** Alpk1 deficiency in haematopoietic cells drives exacerbated colitis and intestinal Th1 responses in *Hh*-infected lymphocyte-replete mice. **a** Experimental schematic of colitis induced by oral infection with *Hh* and systemic blockade of the IL-10 receptor (legend applies to **b–g**). For panels **c**, **e**, and **g**, data shown summarize two independent experiments: steady state (n = 9), *Hh*/αIL-10R-treated *Alpk1*[+/+] and *Alpk1*[+/−] (n = 11), and *Alpk1*[−/−] (n = 6). **b** Representative H&E stained mid-colon cross-sections at steady state and following 12 days of *Hh* infection and αIL-10R treatment. Scale bars = 100 μm. **c** Colon histopathology scores. Only *Hh*-treated groups analyzed by Mann–Whitney test. **d** Colonoscopy-based colitis scores from one of two experiments, the *Hh*-treated groups were analyzed by Mann–Whitney test. Steady state (n = 6); *Hh*/αIL-10R-treated *Alpk1*[+/−] (n = 4); *Hh*/αIL-10R-treated *Alpk1*[−/−] (n = 4). **e** Total live CD45[+] colon lamina propria leucocytes in steady state and colitic mice. *Hh*-treated groups were analyzed by Mann–Whitney test. **f–g** Flow cytometry analysis of cytokine expression by colon lamina propria CD4[+] T cells stimulated with PMA and ionomycin in the presence of brefeldin A. **f** Representative plots of IFN-γ and IL-17A expression. **g** Frequencies of IFN-γ[+] IL-17A[−] and IFN-γ[−] IL-17A[+] cells among total CD4[+] T cells, two *Hh*-treated groups were analyzed by Mann–Whitney test. **h–k** Analysis of *Alpk1*[+/−] and *Alpk1*[−/−] mice at steady state (n = 6), or after 2 weeks of treatment with IL-10R neutralizing antibody (*Alpk1*[+/−] n = 6; *Alpk1*[−/−] n = 4), infection with *Hh* (*Alpk1*[+/−] n = 7; *Alpk1*[−/−] n = 6), or treatment with anti-IL-10R antibody and infection with *Hh* (*Alpk1*[+/−] n = 4; *Alpk1*[−/−] n = 4). Data represent two independent experiments. Bar charts indicate means ± SEM, compared using multiple *t*-tests with Holm-Sidak multiple testing correction. **h** Colonoscopy-based colitis scores. **i** Whole-colon gene expression analysis by RT-qPCR. **j** Representative flow cytometry analysis of colonic CD4[+] T cells at steady state or after *Hh* infection without anti-IL-10R treatment. Top panels depict IFN-γ and IL-17A expression after stimulation with PMA/ionomycin and brefeldin-A; bottom panels depict RORγt and FOXP3 expression in unstimulated cells. **k** Frequencies and absolute numbers of CD4[+] IFN-γ[+] T cells (left two panels). Frequencies of CD4[+] FOXP3[+] and CD4[+] FOXP3[+]RORγt[+] T cells are shown in the right two panels

showing a statistically significant difference, and only 9 > 2-fold (Supplementary Fig. 8b, Supplementary Data 2). This confirms that Alpk1 did not dramatically affect the differentiation of BMDMs in vitro. In contrast, 599 genes were significantly differentially expressed between the genotypes after *Hh* stimulation (adjusted *p*-value <0.05; 67 genes with >2-fold change), including *Il12a* and *Il12b* (Fig. 5d, Supplementary Fig. 8b, Supplementary Data 2). Pathway enrichment and clustering analyses of the top upregulated genes in *Hh*-treated *Alpk1*[−/−] cells revealed enrichment of processes related to IL-12 signalling, regulatory interactions between immune cells (e.g., components of CD200R and CD300 signalling pathways), chemokine signalling, and the extracellular matrix (Fig. 5e, Supplementary Fig. 8c, Supplementary Data 2). Expression of selected genes at the mRNA level was confirmed using qPCR (Fig. 5f).

Precisely how *Hh* triggers TLR2 and downstream IL-12 production in mouse BMDMs is unclear. We found that classical TLR and C-type lectin receptor (CLR) agonists, including LPS, CpG, zymozan, and heat-killed mycobacteria, induced comparable IL-12 expression in wild type and *Alpk1*[−/−] BMDMs (Supplementary Fig. 8d). Assessment of key signalling molecules in the TLR2 pathway, including MAPK kinases and NF-κB revealed no differences between wild type and *Alpk1*[−/−] cells after *Hh* stimulation (Supplementary Fig. 8e).

**Alpk1 expression in human inflammatory bowel disease.** We previously reported that Alpk1 expression is detectable in murine macrophages under steady-state conditions, but can be further induced by stimulation with LPS or *Hh*[3]. IFN-γ stimulation

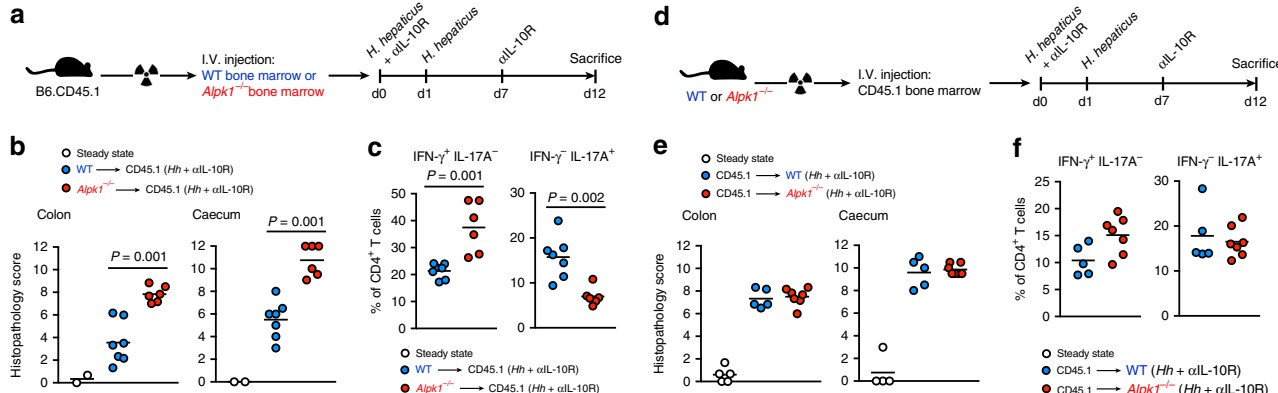

**Fig. 4** Control of colitis by Alpk1 depends on its expression in the hematopoietic compartment. **a–f** Reciprocal bone marrow chimaera experiments in which B6.CD45.1 mice are irradiated and reconstituted with wild type or *Alpk1⁻/⁻* bone marrow **a–c**, or wild type and *Alpk1⁻/⁻* mice are irradiated and reconstituted with wild-type B6.CD45.1 bone marrow **d–f**. After reconstitution, mice are subjected to *Hh* + αIL-10R colitis. The *Hh*-treated groups were compared using Mann–Whitney tests. **a–c** Colon and caecal histopathology scores (**b**) and frequencies of IFN-γ⁺ IL-17A⁻ and IFN-γ⁻ IL-17A⁺ cells among total colonic CD4⁺ T cells (**c**). n = 7 recipients of WT bone marrow, n = 6 recipients of *Alpk1⁻/⁻* bone marrow, and n = 2 steady state. One experiment was performed. **d–f** Colon and caecal histopathology scores (**e**) and frequencies of IFN-γ⁺ IL-17A⁻ and IFN-γ⁻ IL-17A⁺ cells among total colonic CD4⁺ T cells (**f**). n = 5 WT recipients, n = 7 *Alpk1⁻/⁻* recipients, and n = 5 steady state. One experiment was performed

similarly increases *Alpk1* expression in mouse BMDMs (Supplementary Fig. 8f). Consistent with these findings, *ALPK1* mRNA is highly expressed in the inflamed intestinal mucosa of patients with IBD relative to tissue from healthy control donors in three independent cohorts (Fig. 6a). Notably, *ALPK1* expression in IBD tissue was closely correlated with that of Th1-related cytokines (*IFNG*, *IL12A*, *IL12B*, and *CXCL10*) but not Th2 or Th17 cytokines (*IL4*, *IL13*, and *IL17A*) (Fig. 6b). Thus, in addition to regulation by pattern recognition receptors, Alpk1 expression may be induced by IFN-γ signalling as part of a negative feedback mechanism to limit the intensity of IL-12 production and downstream Th1 immunity.

## Discussion

In this study, we have identified Alpk1 as a negative regulator of intestinal inflammation. We demonstrate that Alpk1-deficient mice infected with a pathobiont *Helicobacter hepaticus* (*Hh*) develop an unusually potent Th1 CD4⁺ T-cell response and exacerbated colitis in the absence of IL-10 signalling. Unlike IFN-γ-producing Th1 cells, the frequencies of FOXP3⁺ regulatory and IL-17-producing (Th17) CD4⁺ T cells are decreased in these animals compared to the wild-type controls. Using bone marrow chimera experiments, we also establish that Alpk1 operates in hematopoietic cells to regulate *Hh*-induced pathology and the Th1 response. We show that Alpk1 acts as a checkpoint specifically limiting IL-12 production in mononuclear phagocytes, and that this occurs independently of the classical anti-inflammatory cytokine IL-10.

Our results suggest distinct immunoregulatory roles for Alpk1 and IL-10 in control of *Hh* driven inflammation (Fig. 3). Although *Hh* infection of *Alpk1⁻/⁻* mice resulted in marked skewing of T cell differentiation towards Th1 cells in the colon compared to wild-type mice this was not sufficient to result in severe colitis. By contrast, combined *Hh* infection and IL-10R blockade caused severe disease with high numbers of colonic Th1 and myeloid cells in Alpk1-deficient mice, as compared to wild-type littermates. These results suggest that Alpk1 and IL-10 represent complimentary checkpoints in the Th1 inflammatory response. Alpk1 may be a critical modulator of CD4⁺ T cell differentiation through its suppressive effects on IL-12 production, whereas IL-10 is more important for limiting the magnitude

of *Hh*-driven inflammation through controlling the myeloid response[30].

We have shown that *Hh* stimulation induces elevated IL-12 expression in *Alpk1⁻/⁻* BMDMs, which is blocked by a neutralising TLR2 antibody (Fig. 5b). However, stimulation of BMDMs with a TLR2 ligand Pam3CSK4 or other purified pattern recognition receptor ligands induced comparable IL-12 expression in wild type and *Alpk1⁻/⁻* deficient cells. We speculate that there may be a second, unknown factor under the control of Alpk1 that partners with TLR2 to mediate *Hh* recognition. Another possibility is that Alpk1 regulates phagocytic clearance of *Hh* and signalling pathways linked to this process. Interestingly, Alpk1 was initially shown to regulate apical transport in epithelial cells via interaction with myosin Ia[26]. It is possible that a similar machinery is recruited to aid in *Hh* processing by phagocytes. Deciphering the Alpk1 interactome will be required to investigate this further.

Contrary to the data obtained from in vitro cultured human gastric epithelial cells, in which Alpk1 is essential for IL-8 expression induced by certain bacterial pathogens[12,13], we show that Alpk1 deficiency resulted in elevated cytokine expression in BMDMs treated with *Hh*. This may be explained by a possible cell-type-specific function of Alpk1, as observed for some other inflammatory regulators. For instance, IL-18 signalling in intestinal epithelial cells and lamina propria leucocytes has seemingly opposite effects on intestinal inflammation[31]. Indeed, our bone marrow chimera studies indicate that Alpk1 expression by hematopoietic, but not epithelial cells, is important for control of *Hh*-induced colitis in vivo (Fig. 4). Consistent with this, in vitro analysis showed that Alpk1 can function in macrophages to control *Hh*-driven IL-12 production. However cell type-specific deletion of *Alpk1* will be required to address the context-specific function of Alpk1 in various cell types.

Our results identify *Alpk1* as the first gene in the *Hiccs/Cdcs1* locus with a definitive role in regulating intestinal inflammation. While our data collectively imply that the risk-conferring alleles of *Hiccs/Cdcs1* contain a hypomorphic variant of *Alpk1*, generation of mice with specific point-mutations in *Alpk1* will be required to assess *Alpk1* as the causative gene in the *Hiccs/Cdcs1* locus. The same genomic interval contains *Tifa* (TRAF Interacting Protein With Forkhead Associated Domain), which encodes an adaptor protein involved in TNFR/TLR and NLRP3

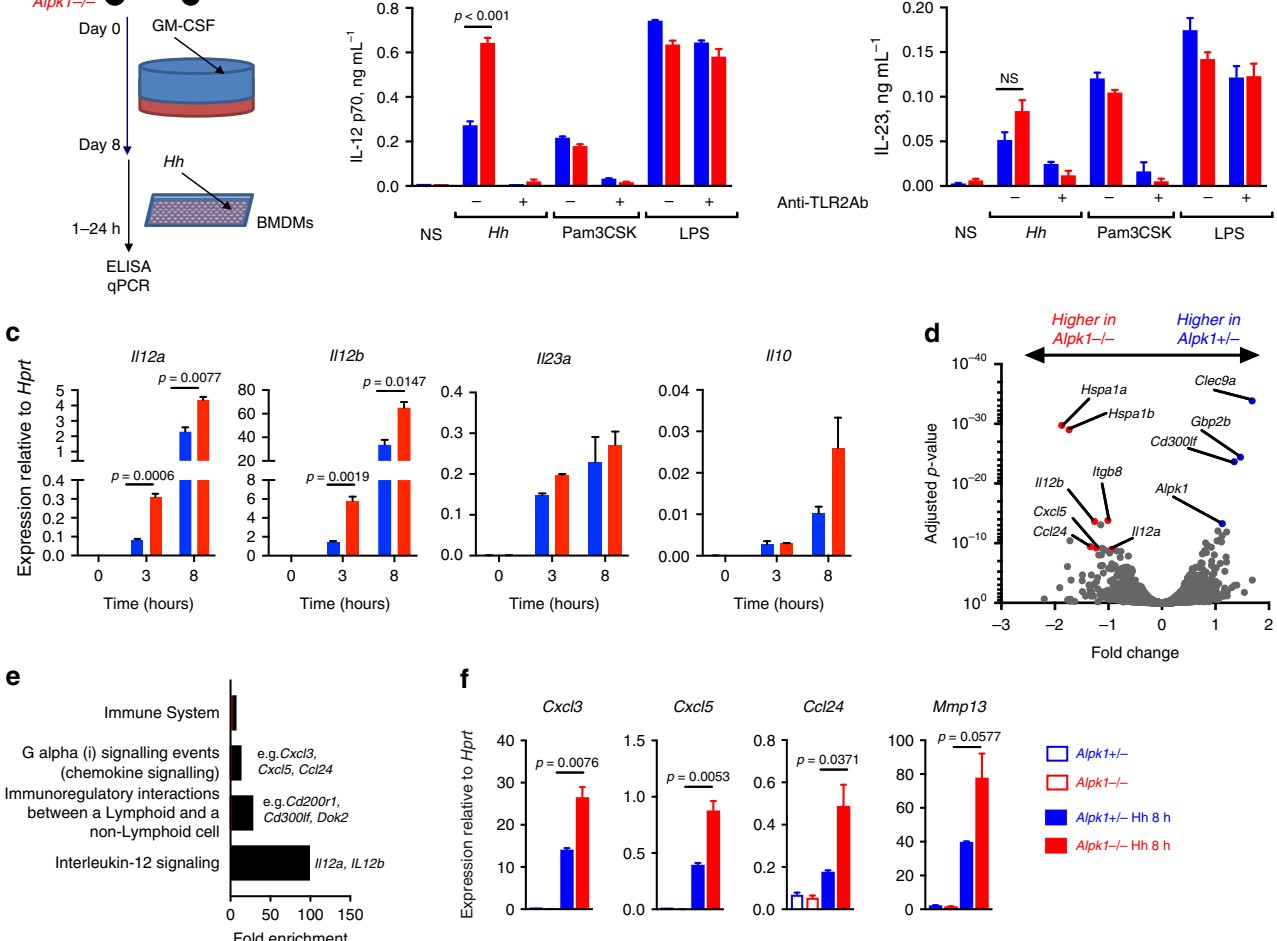

**Fig. 5** Alpk1 controls *Hh*-induced immune gene expression by in vitro differentiated macrophages. **a** Mouse bone marrow-derived macrophages (BMDMs) were differentiated for 8 days in the presence of GM-CSF for subsequent stimulation with *Hh* and TLR agonists. **b** Analysis of IL-12 and IL-23 expression by ELISA. *Alpk1*[+/−] or *Alpk1*[−/−] BMDMs were stimulated with *Hh* (0.5 O.D. mL[−1]), Pam3CSK (100 ng mL[−1]) or LPS (100 ng mL[−1]) in the presence or absence of a neutralizing anti-TLR2 (0.3 μg mL[−1]) antibody. Data shown are from pooled bone marrows ($n = 3$) from one of two independent experiments, data from the *Hh*-treated group were analyzed by two-tailed unpaired *t*-test ($t = 12.99$; df = 4). Bars indicate mean of three replicates ± SEM. **c** Analysis of gene expression by qRT-PCR of BMDMs stimulated with *Hh* (0.5 O.D. mL[−1]) for 3 or 8 h. Data are from pooled bone marrows ($n = 3$) from one of three independent experiments. *P*-values corrected using the Holm-Sidak method are shown. Bars indicate mean of three replicates ± SEM. **d** Volcano plot depicting global gene expression in *Hh*-treated *Alpk1*[+/−] vs *Alpk1*[−/−] BMDMs. Differences in gene expression between genotypes are expressed as log2 fold change (*x*-axis) with adjusted *p*-values (*y*-axis). **e** Enriched pathways in *Alpk1*[−/−] cells based on top differentially expressed genes in *Hh*-treated *Alpk1*[+/−] vs *Alpk1*[−/−] BMDMs. The overrepresentation analysis was carried out using the Reactome pathways[46] and the Panther database[45]. **f** qRT-PCR validation of top differentially expressed genes revealed by RNA-Seq analysis in *Alpk1*[+/−] and *Alpk1*[−/−] BMDMs. Data shown are from pooled bone marrows ($n = 3$) from one of two independent experiments, analysed by unpaired *t*-test. Bars indicate mean of three replicates ± SEM

inflammasome signalling[32,33]. We did not focus on *Tifa* in our studies as there are no non-synonymous nucleotide differences in the *Tifa* coding region and *Tifa* mRNA expression levels are similar in myeloid cells between the colitis susceptible and resistant mouse strains[3]. However, TIFA has been recently implicated in sensing of the bacterial metabolite HBP by human epithelial cells, a process that also requires Alpk1[12,13]. Therefore, involvement of TIFA in Alpk1-dependent pathways in the gut cannot be excluded. Considering the distinct roles of Alpk1 in human epithelial and mouse myeloid cells, generation of TIFA knockout mice is essential to explore this axis further.

SNPs in the *ALPK1* gene have been shown to be associated with inflammatory disease risk in humans[10,11], yet no link has been established between *ALPK1* and inflammatory bowel disease. Taking this into account, we examined *ALPK1* expression in tissue biopsies from patients with IBD and observed elevated

expression compared to healthy individuals. Moreover, Alpk1 expression correlated with the expression of Th1 but not Th2 axis genes in IBD (Fig. 6). These data are in line with our findings that Alpk1 expression in BMDMs is induced by both TLR stimulation[3] and IFN-γ, and therefore may represent a negative feedback loop that constrains intestinal Th1 responses. This concept is consistent with the well-known induction of the negative regulator IL-10 by TLR activation, and elevated IL-10 expression found in inflamed tissues. These human findings necessitate additional work to determine the mechanisms that regulate Alpk1 expression, as well as the spatiotemporal pattern of intestinal Alpk1 expression in healthy, inflamed and dysbiotic settings.

In summary, we have identified *Alpk1* as a critical determinant of murine colitis susceptibility encoded by the *Hiccs* and *Cdcs1* loci. By selectively regulating production of IL-12 downstream of bacterial stimulation, Alpk1 restricts the activation of Th1

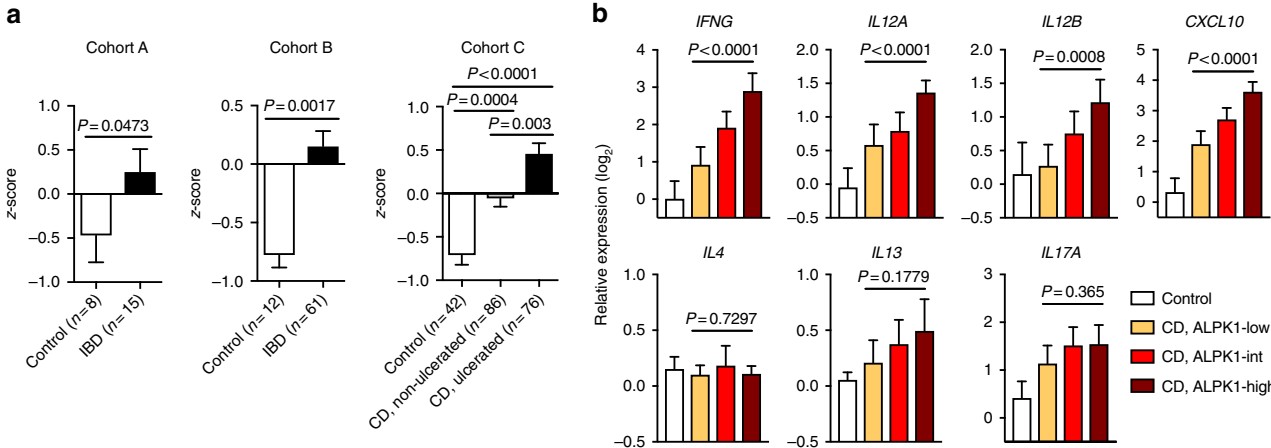

**Fig. 6** Alpk1 expression is elevated in inflamed mucosa of IBD patients and correlates with expression of Th1 signature genes. **a** *ALPK1* mRNA expression in intestinal biopsies of healthy controls or IBD patients from three independent publicly available cohorts described in Materials and Methods (Cohort A, GSE4183; Cohort B, GSE16879; Cohort C, GSE57945). Groups were compared by Mann–Whitney test (cohorts A and B) or Kruskal–Wallis test with Dunn's multiple comparisons tests (cohort C). Bars indicate mean z-scores ± SEM. **b** RNA-seq gene expression analysis of ileal biopsies from healthy control ($n = 42$) and ileal CD patients (Cohort C). CD samples were stratified by relative *ALPK1* expression into three equally sized groups ($n = 54$ per group) and compared using the Kruskal–Wallis test. Bars indicate mean $\log_2$ RPKM values ± 95% CI, normalized to the control mean

immunity in favour of Treg and Th17 responses that maintain homoeostasis. Intriguingly, Alpk1 and IL-10 appear to regulate intestinal homoeostasis via distinct mechanisms, in that Alpk1 controls the nature of intestinal immune responses, while IL-10 controls their magnitude. Thus, loss of Alpk1 function permits aberrant Th1 immunity that, in combination with IL-10 deficiency, causes aggressive and highly destructive colitis (Fig. 7). How Alpk1 influences human pathology, and the precise biochemical mechanism by which it regulates expression of inflammatory factors, are key outstanding questions that should be addressed in future studies.

While this paper was in press, data showing a functional role for Alpk1 in anti-microbial responses were published[34].

## Methods

**CRISPR-assisted generation of ALPK1 knockout mice.** Using the web based tool[35] four sgRNA, two sgRNA flanking either side of exon 10 of mouse *Alpk1* gene (ENSMUSE00001278253) were identified. Deletion of exon 10 generates a frame-shift mutation upstream of the alpha-type protein kinase domain. The guide sequences (sgRNAs) were ordered from Sigma Genosys as sense and antisense oligonucleotides and annealed before individually cloning into the T7 expression vector (kind gift from Sebastian Gerety). Following plasmid linearization, RNA was transcribed using a MEGAshortscript T7 Transcription Kit (Thermo Fisher Scientific) and column purified (RNeasy, Qiagen). The humanised Cas9 protein developed by Mali et al.[36] was modified by replacing the CMV promotor with a dual CAG-T7 promotor cassette (kind gift from Katharina Boroviak), thereby allowing expression in the mouse zygote and in vitro transcription of Cas9 mRNA. The vector was linearised and subjected to mMESSAGE mMACHINE® T7 ULTRA Transcription (Thermo Fisher Scientific) before column purification (RNeasy, Qiagen). 75 ngs of Cas9 mRNA, 9.3ngs each of 4 guides was injected into the cytoplasm of fertilised C57BL6n oocytes and transferred into CBAB6F1/J recipients. All applicable European, national, and institutional guidelines for the care and use of animals were followed. All procedures performed in studies involving animals were in accordance with the ethical standards of the Sanger Institute.

**Mice.** Wild-type C57BL/6J (referred to elsewhere in the manuscript as B6), 129S6/SvEvTac-Rag2$^{tm1Fwa}$ and the congenic strain 129.C3BR17.rag[3] (129.Rag2$^{-/-}$ and 129.Hiccs$^{B6}$.Rag2$^{-/-}$, respectively), C57BL/6.Alpk1$^{em2Wtsi}$ (*Alpk1$^{-/-}$*), B6.SJL-Ptprc$^a$Pepc$^b$/BoyJ (B6.CD45.1), C57BL/6.Rag1$^{-/-}$ (B6. Rag1$^{-/-}$) and C57BL/6. Alpk1$^{-/-}$Rag2$^{-/-}$ (B6. Alpk1$^{-/-}$Rag1$^{-/-}$) mice were bred and maintained under specific pathogen free conditions in an accredited animal facility at the University of Oxford and the experiments were conducted in accordance with the UK Scientific Procedures Act of 1986 under a Project License (PPL) authorized by the UK Home Office Animal Procedures Committee. For experiments involving *Alpk1$^{-/-}$* mice, knockout animals and wild type/heterozygous littermate controls were co-housed. 129SvEvS6.Rag2$^{-/-}$ and R17 congenic strains were bred separately and for

experiments the mice from both strains were co-housed after weaning for at least a month before experiments.

For genotyping of Alpk1 knockout mice the following primers were used: Alpk1_DF4 GGAGGGGGAGAGAAGTTTGT, Alpk1_DR3 GTCCCTATTCCATCCCTTTTG and Alpk1_ER1 GTCAGGAGGCCAATCTCAAA, with the DF4/ER1 PCR detecting the wild-type *Alpk1* allele (190 bp band) and the DF4/DR3 PCR—the Crispr-mutated allele (150 bp band). To ensure maintenance of the Cdcs1 BC-R2 interval during crossing, *Was$^{-/-}$Cdcs1$^{+/+}$* mice were genotyped at the D3Mit49 and D3Mit348 loci using the following primers:

D3Mit49 - FCTTTTCTCGCCCCACTTTC
D3Mit49 - RTCCTTTTAGTTTTTGATCCTCTGG
D3Mit348 - FCATCATGCATACTTTTTTCCTCA
D3Mit348 - RGCCAAATCATTCACAGCAGA

Cg-Tbx21$^{tm1Glm}$Rag2$^{tm1Fwa}$ (T-bet$^{-/-}$Rag2$^{-/-}$, TRUC) mice have been backcrossed to a C57BL/6J (B6) background for at least 10 generations. Congenic TRUC mice carrying the colitogenic Cdcs1$^c$ QTL from C3H/HeJBir mice on a C67BL/6J background[9] were generated by backcrossing BCR3 mice (B6.Cg-Il10$^{tm1Cgn}$Cdcs1(D3Mit11-D3Mit19)/JZtm)[6] to the C57BL/6 TRUC line. The TRUC.Cdcs1$^c$ mice initially contained between 8 and more than 26 Mbp of C3H/HeJBir-derived genomic sequence (D3Mit78-D3Mit291) and were further backcrossed to C57BL/6 TRUC mice to generate additional mutants. All mice were housed in microisolator cages in a specific pathogen-free animal facility at the Harvard T.H. Chan School of Public Health. Studies were performed according to institutional and NIH guidelines for humane animal use. Sulfatrim (Sulfamethoxazole 1 g L$^{-1}$ + Trimethoprim 0.2 g L$^{-1}$; Hitech Pharmacal, Amityville, NY) was added to the drinking water. Mice were genotyped between 2 and 3 weeks, and weaned between 3 and 4 weeks of age.

Both male and female mice were used in approximately equal proportions for all experiments. For antibody blockade experiments, mice were randomized to allow for at least two treatments represented in each cage of animals. Minimum sample size of six animals per experimental group (3 for steady state) was determined based on experience with colitis models.

**Bone marrow chimeras.** For BM chimera experiments with CD45.1 and Alpk1 knockout mice, recipient mice were lethally irradiated (2 × 550 rads/5.5 Gys) before reconstitution with BM from the indicated donor line. BM was aseptically collected from tibia and femur of the respective donor strain, and 5 × 10$^6$ cells were injected into the tail vein of the irradiated recipients. Mice were allowed to reconstitute for 8 wk before being infected with H. hepaticus.

For TRUC mice, 6–7-week-old recipient mice were lethally irradiated with 800 rad (BALB/c TRUC) or 900 rad (C57BL/6 TRUC). Recipient mice were injected in the lateral tail vein with 2 × 10$^6$ donor cells in PBS. Mice were killed 20 weeks after BM reconstitution. Chimerism determined by flow cytometric analysis of peripheral blood using H-2K$^b$ and H-2K$^d$ specific antibodies (Biolegend) was >95% in all animals.

**BMDM cultures.** Extracted bone marrow cells were plated at density of 7 × 10$^6$ cells per 10 cm bacteriological dish and cultured for 7 days in RPMI (Sigma) medium supplemented with Pen-Strep antibiotics (Sigma), 10% FCS (Gibco), 50

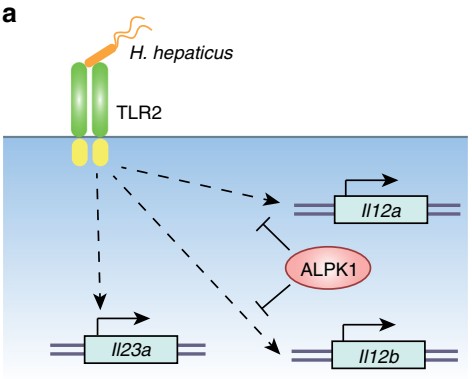

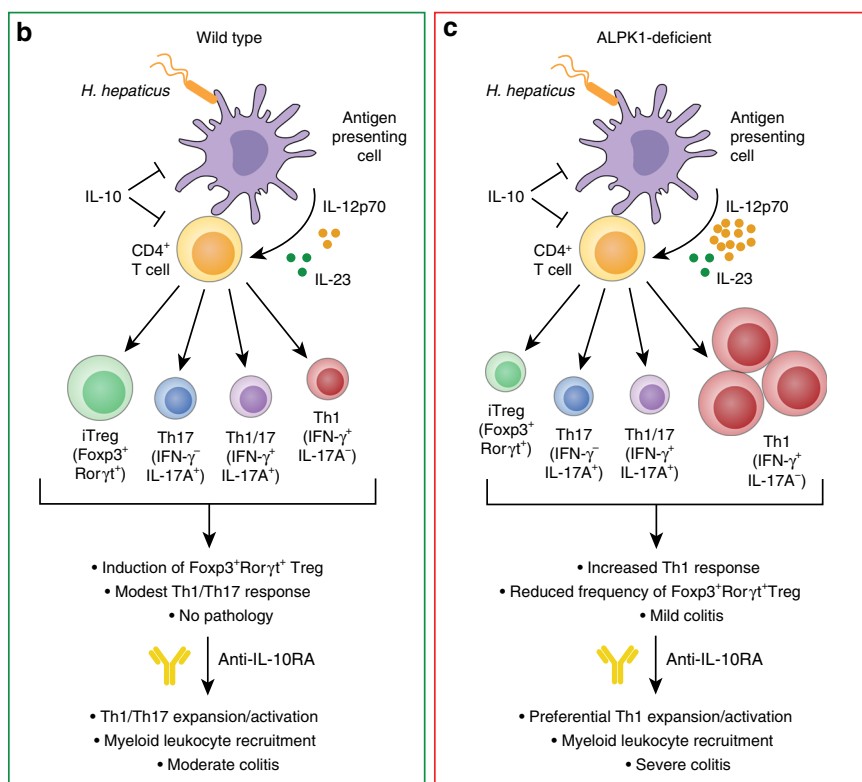

**Fig. 7** Working model for the role of Alpk1 in controlling gut immune homoeostasis. **a** TLR2 signaling in response to *Helicobacter hepaticus* induces expression of *Il12a*, *Il12b*, and *Il23a* in antigen presenting cells (APCs). Alpk1 selectively inhibits expression of *Il12a* and *Il12b*, but does not significantly restrain *Il23a* expression. **b** In wild-type APCs, control of *Il12a* and *Il12b* expression by Alpk1 results in balanced and modest secretion of IL-12p70 and IL-23 in response to *H. hepaticus*. IL-10 acts independently to further control release of pro-inflammatory cytokines. This results in activation of FOXP3+RORγt+ inducible Tregs and modest induction of Th1 and Th17 effector CD4+ T cells, which does not cause detectable pathology or significant alterations in effector cytokine production in the gut. Additional blockade of IL-10 signaling using an anti-IL10R antibody results in robust activation of a mixed population of Th17 and Th1 effector T cells, recruitment and activation of myeloid cells, and moderate colitis. **c** In Alpk1-deficient mice, *H. hepaticus* stimulates potent production of IL-12, which promotes polarized activation of pro-inflammatory Th1 effector T cells at the expense of Tregs and Th17 cells, and mild but detectable inflammation. Following blockade of IL-10R, Th1 cells expand dramatically, inflammatory myeloid populations are recruited to the intestine, and severe colitis ensues. This demonstrates that Alpk1 controls the quality of T cell responses following *Helicobacter* challenge by restraining IL-12 production and Th1 differentiation, thus promoting a balanced Treg/Th17/Th1 response that maintains homoeostasis. In an Alpk1-deficient setting, Th1 differentiation dominates the *Helicobacter*-induced T cell response, but IL-10 acts in parallel to restrain Th1 cell expansion and innate inflammation. Removal of both homoeostatic control mechanisms (Alpk1 and IL-10) allows the rapid onset of Th1-driven colitis and ensuing tissue destruction

μM β-mercaptoethanol (Gibco), GlutaMax$^{TM}$ (Invitrogen) and 20 ng mL$^{-1}$ of murine GM-CSF (Peprotech, #315-03). The medium was topped up after 4 days of culture. On day 8, BMDMs were plated in round bottomed 96-well plates at density of $1.5 \times 10^5$ cells/well and the following day the BMDMs were treated with various microbial stimuli in the presence of 5 ng mL$^{-1}$ recombinant mouse IFNγ (Peprotech, #315-05) for 3 h, 8 h, or 24 h. *Hh* (defrosted bacterial pellets) was used at 0.5 OD/well, ultrapure LPS from *E. coli* O111:B4 at 10 ng mL$^{-1}$ (#tlrl-3pelps),

Pam3CSK4 (#tlrl-pms)—100 ng mL$^{-1}$, zymozan (TLR2/dectin-1 agonist, #tlrl-zyn) —1 μg mL$^{-1}$, ODN1826—0.2 μg mL$^{-1}$ (CpG, TLR9 agonist, #tlrl-1826), HKMT— 10 μg mL$^{-1}$ (Mincle agonist, #tlrl-hkmt-1), neutralising anti-mouse TLR2 antibody (clone C9A12, # mabg-mtlr2)—0.3 μg mL$^{-1}$ (all from Invivogen). Cytokine expression in the culture supernatant was determined using mouse IL-12 Duoset ELISA kit (RnD Systems) and mouse IL-23 ELISA Ready-SET-Go!™ Kit (eBioscience).

**Western blot analysis.** Cell lysates were diluted in the sample buffer (2X, 0.05% bromophenol blue, 20% glycerol, 4% SDS, 0.25 M Tris-HCl, pH 6.8 and 200 mM DTT) and separated by SDS PAGE using precast 4–12% denaturing gels (Invitrogen), immunoblotted and the protein bands were visualized using Amersham ECL detection system (GE Healthcare). Antibodies were from Cell Signalling—pT202/Y204-ERK1/2 (clone D13.14.4E, #4370), pT180/Y182-p38 (clone 3D7, #9215), pT581-MSK1 (#9595), pS473-Akt1 (clone D7F10, #9018), pThr183/Tyr185-JNK (#9251), Sigma—total β-actin (#A1978), Abcam—pS536-RelA (#ab86299).

**T-cell transfer model of colitis.** B6.$Rag1^{-/-}$ mice were injected intraperitoneally (i.p.) with $0.2 \times 10^6$ FACS-sorted naive CD45RB$^{high}$ CD4$^+$ T cells derived from mouse spleens. Mice were monitored bi-weekly for weight loss. 6–8 weeks after the injections, after the weight loss was approaching 20%, mice were killed and the extent of colitis was analysis by FACS and histology.

**αCD40 model of murine colitis.** To induce acute innate colitis, C57BL/6 $Rag1^{-/-}$ mice were injected intraperitoneally with 50 μg of αCD40 IgG2a mAb (clone FGK45, BioXCell, West Lebanon, NH). Weight loss was monitored daily. Two days post αCD40 injection mice were killed to characterise the extent of colitis.

**Hh-driven colitis models and in vivo treatments.** *Helicobacter hepaticus, Hh* NCI-Frederick isolate 1A (strain 51449) was grown on blood agar plates containing 7% laked horse blood (Thermo Scientific) and trimethoprim, vancomycin and polymixin B (all from Oxoid) under microaerophilic conditions. Cultures were expanded for 3–4 days in TSB (Oxoid) containing 10% FCS (Gibco) and the above antibiotics to OD 0.4–0.5 before collection for oral gavaging. Mice were fed $1 \times 10^8$ colony forming units (c.f.u.) of *H. hepaticus* by oral gavage delivered with a 22G curved blunted needle on days 0 and 1 of the experiment (innate model, 129.$Rag2^{-/-}$, $129^{HiccsB6}.Rag2^{-/-}$, C57BL/6 $Rag1^{-/-}$, and C57BL/6 $Alpk1^{-/-}Rag1^{-/-}$ mice). In experiments involving $Alpk1^{-/-}$ mice on the replete $Rag1^{+/+}$ background, 1 mg of an IL-10R blocking antibody (clone 1B1.2) was administered as an intraperitoneal injection once weekly starting at day 0 of *Hh* infection. For in vivo cytokine blockade experiments, mice were injected intraperitoneally with 0.8 mg IL-23R blocking mAb (D. Cua, Merck, Kenilworth, NJ), or anti-IL-12p40 (clone CB17.8.20) or an isotype control antibody (clone GL117) once weekly starting a day before *Hh* infection.

**Scoring of mouse colitis.** Colonoscopy to assess colitis severity was performed and scored according to the methods of Becker et al.[37]. Histological assessment of colitis severity was performed following the established procedures[38]. Briefly, formalin-fixed paraffin-embedded cross-sections of proximal, middle, and distal colon were stained with haematoxylin and eosin and graded on a scale of 0–3 for four parameters: epithelial hyperplasia and goblet cell depletion, leucocyte infiltration, area affected, and features of severe disease activity. Common severity features include crypt abscess formation, submucosal leucocyte infiltration, and interstitial oedema. Scores for each criterion are added to give an overall score of 0–12 per colon section. Data from the three colon regions are then averaged to give an overall score. Scoring was conducted in a blinded fashion and confirmed by an independent blinded observer. Interobserver Pearson correlation coefficients ranged from 0.8 to 0.9. Photomicrographs of H&E stained colon sections were taken with a Coolscope Slide Scanner (Nikon). For TRUC mice scoring, the histological parameters were mononuclear cell infiltration, polymorphonuclear cell infiltration, epithelial hyperplasia, and epithelial injury were scored as absent (0), mild (1), moderate (2), or severe (3), giving a total score of 0–12[9].

**Mouse colon tissue preparation and cell isolation.** LPLs were isolated following established procedures[39]. Briefly, mouse colons were washed with EDTA to remove epithelium and digested with collagenase VIII to liberate cell populations. Tissue digests were separated by centrifugation on a 40%/80% Percoll (Sigma) gradient. Cells at the 40%/80% interface were collected as the lamina propria leucocyte enriched fraction.

**Colon explant cultures.** 3 mm segments isolated from a middle part of mouse colon were cultured overnight in RPMI media supplemented with Pen-strep antibiotics (Sigma), 10% FCS (Gibco) and 50 μM β-mercaptoethanol (Gibco). IFNγ was quantified in the supernatant by enzyme-linked immunosorbent assay (ELISA, R&D Systems, UK) and normalized to explant weight (mg of tissue).

**Quantitation of H. hepaticus using real-time PCR.** DNA was purified from caecal contents taken from *H. hepaticus*–infected mice using the DNA Stool kit (Qiagen). *H. hepaticus* DNA was determined using a Q-PCR method based on the *cdtB* gene[40].

**RNA extraction, cDNA synthesis, and qPCR.** Tissues were disrupted using lysis beads and a homogenizer unit (Precellys, UK) in the RLT buffer (Qiagen, UK). Sorted or cultured cells were lysed directly in the RLT buffer and homogenized by pipetting. RNA was isolated using RNEasy Mini or Micro kits (Qiagen, UK) followed by reverse transcription using random primers (Applied Biosystems, UK). Quantitative PCR (qPCR) was performed using Taqman assays (Applied Biosystems) and PrecisionPlus Mastermix (Primer Design, UK) on a ViiA7 384-well real-time PCR detection system (Applied Biosystems). All expression levels were normalized to an internal house-keeping gene *Hprt* and calculated as $2^{(CTHprt-CTgene)}$.

**Microarray analysis.** Expression profiles of colon tissues from 129.$Rag2^{-/-}$ and $129.^{HiccsB6}.Rag2^{-/-}$ were obtained using Illumina MouseWG-6-V2 microarrays ($n = 4$ for each condition). Array signal intensities were background adjusted, transformed using the variance-stabilizing transformation and quantile normalized using Lumi[41] from R/Bioconductor. Probes were retained if they were expressed significantly above background levels in at least four samples. This resulted in the analysis of 20,997 probes representing 15,443 genes. Differential expression analysis was performed using the empirical Bayes method in LIMMA[42]. Significance was defined as a Benjamini-Hochberg adjusted $P$-value <0.05.

**RNA sequencing analysis.** PolyA-selected RNA from in vitro differentiated macrophages (Fig. 5) from pooled bone marrows from three mice (split into three technical replicates per condition) was used to prepare cDNA libraries (in-house dUTP protocol) that were subject to next-generation sequencing (Illumina HiSeq4000; 75 bp; minimum of 50M read pairs per sample). Mapping was performed using HISAT2[43] (two pass strategy to incorporate novel splice-sites) together with an index built from the mouse genome (mm10) and Ensembl transcript annotations (version 88). Differential expression analysis was performed using read counts (quantitated with featureCounts[44]) and the DESeq2 algorithm[45] (local fit). $P$-values were adjusted for multiple-testing using the Benjamini-Hochberg correction.

**Pathways and gene set enrichment analysis.** We tested for the enrichment of pathways that were differentially upregulated at day 2 of *Hh* infection in 129.$Rag2^{-/-}$ and 129$^{HiccsB6}.Rag2^{-/-}$ mice. Gene set enrichment analysis (GSEA) was conducted using the GSEA desktop program; Hallmark gene sets were downloaded from the Molecular Signatures Database (http://software.broadinstitute.org/gsea/msigdb/collections.jsp). For gene ontology enrichment analysis (focusing on GO_ImmuneSystemProcess and REACTOME terms), the top 100 most significantly differentially expressed genes in 129.$Rag2^{-/-}$ vs 129$^{HiccsB6}.Rag2^{-/-}$ mice after 2 days of *Hh* stim were analysed using ClueGO v2.1.7 with a minimum kappa score threshold of 0.5 and right-sided hypergeometric test with Bonferroni step-down correction to identify significant terms. For RNA-Seq, top significant (adj. $p$-value <0.01) genes differentially expressed in $Alpk1^{+/-}$ and $Alpk1^{-/-}$ BMDMs were analyzed using the online tool Panther[46] and Reactome pathways[47] as a source of terms for pathway enrichment analysis.

**Flow cytometry and cell sorting.** Mouse cells were stained with combinations of the following monoclonal antibodies, with dilutions/catalogue numbers indicated in brackets, according to manufacturer protocols: B220-PerCP (1/400; RA33-6B2), CCR7-PE (1/100; 4B12), CD4-BV605 (1/200; RM4-5), CD11b-PerCP-Cy5.5 (1/400; M1/70), CD11b-BV510 (1/300; M1/70), CD11c-efluor450 (1/200; N418), CD11c-BV605 (1/200; N418), CD11c-PerCP-Cy5.5 (1/400; N418), CD19-biotin (1/300; 6D5), CD25-PE-Cy7 (1/200; PC61.5), CD40-APC (1/150; 1C10), CD44-BV510 (1/150; IM7), CD45-BV650 (1/400; 30-F11), CD45-BV510 (1/400; 30-F11), CD45-AF700 (1/300; 30-F11), CD45.1-PE-Cy7 (1/200; A20), CD45.2-AF700(1/400; 104), CD45RB-APC (1/150; C363.16A), CD62L-PE (1/150; MEL-14), CD86-FITC (1/150; GL1), CD115-PE-Cy7 (1/100; AFS98), CD117-AF700 (1/150; ACK2), CD135-biotin (1/150; A2F10), F4/80-BV421 (1/150; BM8), FOXP3-PE-TxR (1/150; FJK-16s), Gr1-FITC (1/400; RB6-8C5), IFNγ-ef450 (1/500; XMG1.2), TNF-PE-Cy7 (1/200; MP6-XT22), TNF-AF700 (MP6-XT22), IL-17A-FITC (1/150; TC11-18H10.1), IL-17A-BV605 (1/150; TC11-18H10.1), IL-22-PE (1/150; 1H8PWSR), Ly6C-PE-Cy7 (1/300; HK1.4), MHCII-AF700 (1/200; M5/114.15.2), MHCII-BV785 (M5/114.15.2), RORγT-PE (1/400; Q31-378), Siglec-F-PE (1/200; E50-2440), streptavidin-PE-CF594 (1/500; SP34-2), Thy1.2-PE-Cy7 (1/800; 53-2.1) and TCRβ-BV510 (1/100; H57-597). All antibodies were from eBioscience (UK), Biolegend (UK), Becton Dickinson (UK), or R&D Systems (UK). Dead cells were excluded using efluor-780 fixable viability dye (eBioscience). Samples were acquired on FACS LSR Fortessa and FACS LSRII flow cytometers (Becton Dickinson). Cell sorting was performed using a FACS ARIA III (Becton Dickinson). Data were analyzed using FlowJo (Tree Star, USA). For intracellular cytokine staining, cells were restimulated with PMA (10 ng mL$^{-1}$; Sigma-Aldrich), ionomycin (1 μg mL$^{-1}$; Sigma-Aldrich), and 5 μg mL$^{-1}$ brefeldin A (Sigma-Aldrich). After 3 h, cells were stained with fixable viability dye and surface markers, fixed with BD FACS Lysing Solution (Becton Dickinson, UK), and stained for intracellular cytokines in permeabilization buffer containing 0.05% saponin (Sigma-Aldrich). For staining FOXP3, cells were stained with fixable viability dye and surface markers prior to fixation and permeabilization using the FOXP3 staining buffer kit (eBioscience) according to manufacturer instructions. Gating strategy employed in the study is shown on Supplementary Fig. 9. Boolean analysis of cytokine expression by CD4$^+$ T cells was performed using SPICE with default parameters[48].

**RNAScope.** $Alpk1^{+/-}Rag1^{-/-}$ and $Alpk1^{-/-}Rag1^{-/-}$ mice were untreated ($n = 4$) or infected with *Helicobacter hepaticus* by oral gavage with $1 \times 10^8$ CFU ($n = 5$) and killed after 2 days. Formalin-fixed paraffin-embedded tissues from proximal colons were sectioned at 5 μm and collected onto Superfrost glass slides. Tissue sections were dewaxed in xylene and rehydrated through graded alcohol to water. Antigens were retrieved by boiling the tissue sections in target-retrieval buffer provided by ACD, following the manufacturer's instructions. For detection of mouse *Il12b* and *Alpk1* mRNA, the RNAScope Multiplex Fluorescent Reagent kit v2 (ACD Europe SRL) was used. For *Alpk1* mRNA detection, the probe *Alpk1* was custom made to span all exons with the exception of exon 10, which is excised in Alpk1 KO animals. Briefly, paraffin sections were freshly cut, dried for 1 h at 60 °C and dewaxed before mild unmasking with Target Retrieval buffer and protease, as per the manufacturer's instructions. Pretreated sections were hybridized with specific probes to *Il12b* and *Alpk1*, as well as *Ppib* and *Polr2a* (positive control) and irrelevant probe to *dapb* as a negative control. After hybridization signal amplification, TSA Plus fluorophores (PerkinElmer) were reconstituted and used to develop individual channel signal (*Il12b* in C3, *Alpk1* in C5). Nuclei were stained with DAPI and sections were cover-slipped with Prolong Gold Antifade mounting medium (Thermo Fisher). Images were acquired using the A1 Zeiss Axioscope (Carl Zeiss™) at ×40 and ×100 magnifications. For final presentation, images were enhanced using Adobe Photoshop to increase contrast between the red (*Alpk1*) and yellow (*Il12b*) signals to background DAPI signal. All evaluations were performed in a blinded manner. For *Il12b* mRNA quantification, ×100 magnification images were used to count *Il12b* (yellow) dots in *Alpk1*-expressing cells (red dots). A minimum of 10 individual cells per mouse and per condition was used for quantification. The total number of cells expressing *Il12b* and *Alpk1* in proximal colon was computed using ×40 magnification images.

**Analysis of patient cohorts.** Publicly available whole-transcriptome data from intestinal biopsies of healthy controls and patients with IBD were downloaded from the Gene Expression Omnibus web portal. The following datasets were analysed: GSE4183[49] (cohort A; $n = 8$ control vs $n = 15$ IBD; assessed by Mann–Whitney test), GSE16879[50] (cohort B; $n = 12$ control vs $n = 61$ IBD; assessed by Mann–Whitney test), and GSE57945[51] (cohort C; assessed by Kruskal–Wallis with Dunn's multiple comparisons test).

**Statistical analysis.** Unless otherwise indicated, all bar charts represent means ± S. E.M. Statistical tests were two-sided and specified in figure legends. Differences were considered to be significant when $p < 0.05$. The non-parametric Mann–Whitney test was used for comparing pathology scores, FACS, ELISA, and Q-PCR data in *Hh*-treated groups of mice. Histopathology scores along the length of the large intestine were compared using two-way ANOVA using the Prism software (Graphpad). Where possible, the groups being compared are indicated by a connecting line in figures.

## Data availability

All analysed high-throughput datasets generated during this study are included in Supplementary Data 1, 2. The raw datasets are available in the ArrayExpress repository, accession numbers are E-MTAB-7075 and E-MTAB-7076. All other data are available upon request from the author.

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

## Acknowledgements

We thank the Wellcome Trust Sanger Institute Mouse Genetics Project (Sanger MGP) and the Infection and Immunity Immunophenotyping (3i) consortium for providing the *Alpk1⁻/⁻* mouse. Funding and associated primary phenotypic information may be found at www.sanger.ac.uk/mouseportal. We would like to thank Dr. Dan Cua (Merck) for providing the blocking anti-IL-23R antibody. We would like to thank the staff of our animal houses and histology and FACS facilities for their assistance. This work was supported by grants from the National Institute of Diabetes and Digestive and Kidney Diseases (NIDDK) DK106311 and the Crohn's and Colitis Foundation CDA 352644 to J.A.G.; grants from NIDDK—DK034854, the Helmsley Charitable Trust and the Wolpow Family Chair in IBD Treatment and Research, and the Translational Investigator Service to SBS; NIH grant R03 AR066357, grants from the Harvard Digestive Disease Center and the Harvard Institute of Translational Immunology/Helmsley Trust to J.E. and grants from the Wellcome Trust UK (095688/Z/11/Z) and European Research Council (Advanced Grant Ares(2013)3687660) to F.P. F.F. was supported by Cancer Research UK (OCRC-DPhil13-FF). N.R.W. was supported by an Irvington Institute Postdoctoral Fellowship (Cancer Research Institute).

## Author contributions

G.R., N.R.W., J.A.G., J.E., B.H.H., S.B.S. and F.P. conceived and designed the study, G.R., N.R.W., F.F., S.C., S.J.B., C.P., A.C., A.V.-J., L.B., J.A.G. and J.E. performed experiments, G.R., N.R.W., F.F., N.E.I., S.N.S., J.A.G., J.E., B.H.H., S.B.S. and F.P. analysed and interpreted data, G.R., N.R.W., and F.P. wrote the manuscript, X.L., S.M., B.H.H. and S.B.S. and F.P. supervised the study. All authors read and approved the manuscript.

## Additional information

**Competing interests:** The authors declare no competing interests.

