## [Peer Review File · Nature Communications]

REVIEWERS' COMMENTS:

Reviewer #1 (Remarks to the Author):

In their manuscript "Alpha kinase 1 controls intestinal inflammation by suppressing the IL-12/Th1 axis", Ryzhakov et al. identify alpha kinase 1 (Alpk1) a gene located in the Hiccs locus as a regulator of intestinal inflammation in response to *Helicobacter hepaticus* infection. In comparing the congenic locus *Cdcs1* associated with HeJBir mice to the Hiccs locus associated with disease susceptibility in 129 mice, they propose that the Alpk1 gene is the responsible element. Interestingly, they note that Alpk1 in the 129 strain (susceptible) differs by 17 nsSNPs from the B6 strain (Resistant). The authors generate an Alpk1 deficient animal on the resistant background (B6) and demonstrate that loss of Alpk1 exacerbates IL-12/IL-23 associated intestinal inflammation. Specifically Alpk1 deficiency in the hematopoietic compartment leads to elevated Th1 responses and increase susceptibility to colitis. The authors note that Alpk1 is highly expressed in macrophages and dendritic cells and consistent with this, bone marrow macrophages lacking Alpk1 produce high levels of IL-12 following infection with *Helicobacter hepaticus*. They do not see this however in other models that lack a strong bacterial signal such as adoptive transfer of T cells into Rag deficient mice. Mechanistically, they suggest that ALPK1 is restraining TLR2 dependent IL12A production. Overall the study is well conceived, properly executed and contains an extensive amount of data.

Major Concern

The authors state "In response to Hh, Alpk1^{-/-} macrophages produce abnormally high amounts of IL-12, but not IL-23, in a toll-like receptor 2 (TLR2)-dependent manner." To claim TLR2 dependency is an overstatement as stimulation with a cognate TLR2 ligand (Pam3CSK4) induced comparable levels of IL12 in Alpk1^{-/-} vs Alpk1^{+/-} macrophages and IL12 expression is not detectable after treatment with TLR2 blocking antibody in Hh model.

Reviewer #2 (Remarks to the Author):

The manuscript "Alpha kinase 1 controls intestinal inflammation by suppressing the IL-12/Th1 axis" by Ryzhakov et al. builds on a previous version submitted to Nature that was reviewed by this referee. The authors identified Alpk1 as a likely regulator of intestinal inflammation in rodent models, mediating this effect through the hematopoietic compartment. As already depicted previously, the authors have elegantly delineated the role of this gene located within a major colitogenic genetic locus by functional studies utilizing Alpk1-deficient mice. This is the first report showing a correlation of the function of a specific gene within the mouse major modifier locus for colitis susceptibility with the function of the susceptibility allele previously defined by QTL analyses. Though the genetic proof that Alpk1 is the causative gene within the Hiccs and the *Cdcs1.3* locus is still missing, the findings presented are of high importance for the field because they point to dysregulation of innate immune mechanisms in the hematopoietic compartment as a causative factor for (experimental) IBD.

The results depicted in the current manuscript are essentially identical to the previous version. However, the manuscript benefits from a more comprehensive discussion, e.g. clearly depicting the limitations (including the lack of the genetic proof of the QTL-gene as well as the lack of the mechanism of action), and from the extended explanation of materials and methods. In addition, all minor issues previously raised by this reviewer have been addressed.

In my eyes, there are only few minor issues the manuscript would further profit from: Mouse genes and genetic loci should be written italics (e.g., *Alpk1* throughout the whole manuscript).

Line 335 ff: The substrain of wildtype C57BL/6 mice should be added as should the full nomenclature of Rag2^{-/-} mice.

Though statistics has much improved in the main body of the manuscript, it should be revised similarly in the suppl. Figures.

Response to reviewers

“Alpha kinase 1 controls intestinal inflammation by suppressing the IL-12/Th1 axis” by Ryzhakov et al.

Dear Editor,

We would like to thank the reviewers for their careful critique of our manuscript. Please find below our point-to-point response to their comments. Reviewer comments are in bold, while our responses are in plain text.

Reviewer #1 (Remarks to the Author):

In their manuscript “Alpha kinase 1 controls intestinal inflammation by suppressing 1 the IL-12/Th1 axis”, Ryzhakov et al. identify alpha kinase 1 (Alpk1) a gene located in the Hiccs locus as a regulator of intestinal inflammation in response to *Helicobacter hepaticus* infection. In comparing the congenic locus *Cdcs1* associated with HeJBir mice to the Hiccs locus associated with disease susceptibility in 129 mice, they propose that the Alpk1 gene is the responsible element. Interestingly, they note that Alpk1 in the 129 strain (susceptible) differs by 17 nsSNPs from the B6 strain (Resistant). The authors generate an Alpk1 deficient animal on the resistant background (B6) and demonstrate that loss of Alpk1 exacerbates IL-12/IL-23 associated intestinal inflammation. Specifically Alpk1 deficiency in the hematopoietic compartment leads to elevated Th1 responses and increase susceptibility to colitis. The authors note that Alpk1 is highly expressed in macrophages and dendritic cells and consistent with this, bone marrow macrophages lacking Alpk1 produce high levels of IL-12 following infection with *Helicobacter hepaticus*. They do not see this however in other models that lack a strong bacterial signal such as adoptive transfer of T cells into Rag deficient mice. Mechanistically, they suggest that ALPK1 is restraining TLR2 dependent IL12A production. Overall the study is well conceived, properly executed and contains an extensive amount of data.

Major Concern

The authors state “In response to Hh, Alpk1^{-/-} macrophages produce abnormally high amounts of IL-12, but not IL-23, in a toll-like receptor 2 (TLR2)-dependent manner.” To claim TLR2 dependency is an overstatement as stimulation with a cognate TLR2 ligand (Pam3CSK4) induced comparable levels of IL12 in Alpk1^{-/-} vs Alpk1^{+/-} macrophages and IL12 expression is not detectable after treatment with TLR2 blocking antibody in Hh model.

We have removed this claim and altered the discussion as suggested by the reviewer.

Reviewer #2 (Remarks to the Author):

The manuscript “Alpha kinase 1 controls intestinal inflammation by suppressing the IL-12/Th1 axis” by Ryzhakov et al. builds on a previous version submitted to Nature that was reviewed by this referee. The authors identified Alpk1 as a likely regulator of intestinal inflammation in rodent models, mediating this effect through the hematopoietic compartment. As already depicted previously, the authors have elegantly delineated the role of this gene located within a major colitogenic genetic locus by functional studies utilizing Alpk1-deficient mice. This is the first report showing a correlation of the function of a specific gene within the mouse major modifier locus for colitis susceptibility with the function of the susceptibility allele previously defined by QTL analyses. Though the genetic proof that Alpk1 is the causative gene within the Hiccs and the *Cdcs1.3* locus is still missing, the findings presented are of high importance for the field because they point to dysregulation of innate immune mechanisms in the hematopoietic compartment as a causative factor for (experimental) IBD.

The results depicted in the current manuscript are essentially identical to the previous version. However, the manuscript benefits from a more comprehensive discussion, e.g. clearly depicting the limitations (including the lack of the genetic proof of the QTL-gene as well as the lack of the mechanism of action), and from the extended explanation of materials and methods. In addition, all minor issues previously raised by this reviewer have been addressed.

In my eyes, there are only few minor issues the manuscript would further profit from:

Mouse genes and genetic loci should be written italics (e.g., *Alpk1* throughout the whole manuscript).

Line 335 ff: The substrain of wildtype C57BL/6 mice should be added as should the full nomenclature of Rag2^{-/-} mice.

This has now been amended.

Though statistics has much improved in the main body of the manuscript, it should be revised similarly in the suppl. Figures.

The statistics have been checked within supplementary figures, the missing df and t values have now been added to Supplementary Figure 8d legend.